# CHID1 Is a Novel Prognostic Marker of Non-Small Cell Lung Cancer

**DOI:** 10.3390/ijms22010450

**Published:** 2021-01-05

**Authors:** Olga V. Kovaleva, Madina A. Rashidova, Daria V. Samoilova, Polina A. Podlesnaya, Rasul M. Tabiev, Valeria V. Mochalnikova, Alexei Gratchev

**Affiliations:** 1N.N. Blokhin National Medical Research Center of Oncology, Kashirskoye Sh. 24, 115478 Moscow, Russia; ovkovaleva@gmail.com (O.V.K.); madina.211@mail.ru (M.A.R.); dashasam@mail.ru (D.V.S.); polina.pod@yandex.ru (P.A.P.); tabievrasul@gmail.com (R.M.T.); mochalnikova70@yandex.ru (V.V.M.); 2Moscow State Academy of Veterinary Medicine and Biotechnology—MVA named after K.I. Scriabin, 23 Academika Scriabina St., 109472 Moscow, Russia

**Keywords:** lung cancer, stroma, macrophage, chitinase-like protein, prognosis

## Abstract

There is an urgent need for identification of new prognostic markers and therapeutic targets for non-small cell lung cancer (NSCLC). In this study, we evaluated immune cells markers in 100 NSCLC specimens. Immunohistochemical analysis revealed no prognostic value for the markers studied, except CD163 and CD206. At the same time, macrophage markers iNOS and CHID1 were found to be expressed in tumor cells and associated with prognosis. We showed that high iNOS expression is a marker of favorable prognosis for squamous cell lung carcinoma (SCC), and NSCLC in general. Similarly, high CHID1 expression is a marker of good prognosis in adenocarcinoma and in NSCLC in general. Analysis of prognostic significance of a high CHID1/iNOS expression combination showed favorable prognosis with 20 months overall survival of patients from the low CHID1/iNOS expression group. For the first time, we demonstrated that CHID1 can be expressed by NSCLC cells and its high expression is a marker of good prognosis for adenocarcinoma and NSCLC in general. At the same time, high expression of iNOS in tumor cells is a marker of good prognosis in SCC. When used in combination, CHID1 and iNOS show a very good prognostic capacity for NSCLC. We suggest that in the case of lung cancer, tumor-associated macrophages are likely ineffective as a therapeutic target. At the same time, macrophage markers expressed by tumor cells may be considered as targets for anti-tumor therapy or, as in the case of CHID1, as potential anti-tumor agents.

## 1. Introduction

Non-small cell lung cancer (NSCLC) is one of the most frequent malignancies worldwide. NSCLC is characterized by poor prognosis and remains a difficult-to-manage disease. The key problems of NSCLC are late diagnosis and lack of reliable prognostic criteria. Many patients die within a year after diagnosis, since the disease is revealed at an advanced stage. Low antigenicity and high heterogeneity lead to passive immune escape. The use of anti-PD-1/PD-L1 and anti-CTLA-4 immune checkpoint inhibitors have entered clinical practice and have revolutionized the therapeutic treatment of this pathology in recent years. However, not all cases of lung cancer are amenable to immunotherapy and, according to statistics, disease-free survival is often less than a year. Today, the search for new biological markers of lung cancer is an extremely topical issue. One of the search directions is the tumor microenvironment, the components of which affect the sensitivity of the tumor to immunotherapy.

It is known that a solid tumor is composed not only of tumor cells, but also of supporting stroma that comprises fibroblasts, endothelial cells, and inflammatory infiltrate. Immune cells of inflammatory infiltrate may constitute up to 40% of total tumor mass, they include macrophages, lymphocytes, and T- and B-cells [1]. These cells produce various cytokines and growth factors that significantly influence tumor development. Tumor cells, in turn, attract immune cells and drive the immune cells’ differentiation and activation [2]. Immunosuppressive properties of tumor stroma not only contribute to its development, but also represent a serious problem for effective immunotherapy. The cells responsible for this feature of tumor stroma comprise several macrophage populations (M2), regulatory T-cells (FoxP3+), and myeloid-derived suppressor cells (MDSC) [3]. There is an urgent need for identification of new prognostic markers for NSCLC.

Macrophages are the main part of tumor immune infiltrate and can represent as much as 50% of its mass [2]. Macrophages are usually classified into M1 (anti-tumor) or M2 (tumor supporting) phenotypes [4]. Though this concept is being re-considered now, macrophages that do not demonstrate pro-inflammatory properties, independent of the type of the stimulus, are considered to be M2. These macrophages express various molecular markers that are used for their detection in various pathological situations. There is no consensus about the set of markers that should be used for definition of a tumor associated macrophage (TAM) phenotype. The most frequently used marker is CD68 as a general macrophage marker, which is used in combination with CD163, CD204, or CD206 to define M2 and with HLA-DR and iNOS, to define M1 [5]. There are several additional M2 markers that have been described that have a diagnostic or prognostic potential with emerging roles in the characterization of TAMs. Among those is the chitinase domain containing protein 1 CHID1 (GL008, SI-CLP) [6]. CHID1 is a member of the Glyco_18 domain-containing proteins family, expressed by type 2 macrophages upon their stimulation with interleukin (IL)-4 and glucocorticoids [6,7]. Expression of CHID1 is documented for TAMs in various tumors, but its role in tumor development remains unclear. Similarly to CHID1, expression of iNOS is documented for TAMs and tumor cells in different types of tumors, including NSCLC, but there is no clear association to prognosis established [8]. On one hand, iNOS expressing macrophages should have proinflammatory phenotype and exhibit anti-tumor activity, on the other hand iNOS may induce immunosuppressive activity of MDSC [8].

In the current study, we performed analysis of tumor stroma cells and tumor stroma molecular markers in 100 NSCLC surgical specimens. Association of TAMs and tumor infiltrating lymphocytes (TILs) to prognosis and clinicopathological features was evaluated. Expression of two TAM markers, iNOS and CHID1, was also analyzed in tumor cells, and their association to prognosis was established.

## 2. Results

### 2.1. Immunohistochemical Staining

To study immune cell markers and their association with clinicopathological properties of NSCLC, we collected formalin-fixed, paraffin-embedded (FFPE) samples from 100 patients with a median age of 60.5 years. Of these patients, 51% had primary adenocarcinoma at the time of diagnosis and 49% had a squamous cell carcinoma. The median follow-up for living patients was 29 months (range, 2–104 months). Overall survival (OS) was defined as the interval between surgery and death or between surgery and the last follow-up for surviving patients. Among the 87 patients who were recruited, 43 (49.0%) died, and 44 (51.0%) remained alive during the follow-up period.

The tumor samples were stained for CD68 (total macrophage marker), CD163, CD206, CD204, CHID1 (M2 markers), inducible nitric oxide synthase (iNOS), IDO1 (M1 marker), FoxP3 (Treg marker), CD3 (mature T-cells marker), CD8, and PD-L1. To determine the prognostic effect of these markers, their association with clinicopathologic characteristics was analyzed. Survival analysis was performed for the entire group of patients, separately for adenocarcinoma and squamous cell carcinoma, and for groups comprising stages I-II and III-IV independent of the histotype. Patients for whom no survival information was available and those who died within one month after surgery were removed from the analysis.

### 2.2. T-Cells

Immunohistochemical staining revealed CD3+ T-cells in all samples studied and CD8+ T-cells in 98% of cases. The cells were found in both stroma and tumor nests. We divided the cases into high and low density of TILs groups according to their number and evaluated possible correlations between their density and clinicopathological parameters, including histological type, tumor location, grade, tumor size, nodal status, and clinical stages (Figure 1A, Appendix A). Higher numbers of CD3+ and CD8+ cells were found to be associated with early stages of the disease. Interestingly, high number of CD8+ cells at early stages is an indicator of poor prognosis (HR = 3.152, *p* = 0.0262), while at later stages it has no prognostic value (Figure 1B).

Samples stained with FoxP3, a marker of regulatory T-cells (Treg), were divided into 4 groups: 0 (no positive cells), 1 (1–5 positive cells), 2 (6–25 positive cells), and 3 (>25 positive cells) per high-power field (HPF). No FoxP3 expression was found in 24% of samples. For the analysis of prognostic value, the samples were divided into two groups: low expression (0–1) and high expression (2–3). Results are presented in Figure 1. Analysis of FoxP3 showed statistically significant association of the number of FoxP3+ Treg with histotype of the tumor (Figure 1A). An increased amount of Treg was found in squamous cell lung carcinoma. However, no correlation with disease prognosis was found (Figure 1B).

### 2.3. Macrophages

We used CD68 and CD163 as the markers of macrophages and M2 macrophages, respectively. As additional M2 markers, CD206, CD204, and CHID1 were used. Immunohistochemical staining for CD68, CD163, CD206, CD204, and CHID1 revealed membrane and cytoplasmic staining of macrophages (Figure 2).

We found CD68+, CD163+, CD206+, CD204+, and CHID1+ TAMs mainly distributed in the tumor stroma and tumor nest. No association of macrophage density to clinicopathological characteristics was found, except that the number of CD204 positive cells was associated with tumor differentiation (Figure 3, Appendix A). Namely, highly differentiated tumors more often contained more CD204 + cells than lowly differentiated ones.

For survival analysis, the samples were divided into high and low density of macrophages groups. Both TAMs in stroma and tumor nests were counted. We studied the association of CD68+, CD163+, CD206+, CD204+, and CHID1+ TAMs. Obtained results are presented on Figure 4.

Survival analysis demonstrated that low numbers of CD163+ macrophages are a factor of good prognosis for squamous cell NSCLC (HR = 0.3438, *p* = 0.0302) and NSCLC in general (HR = 0.5008, *p* = 0.0392), while high numbers of CD204+ macrophages are a good prognosis indicator for the early stages of NSCLC (HR = 0.3187, *p* = 0.0451) (Figure 4).

### 2.4. Macrophage Markers in Tumor Cells

The tumor samples were stained for M2 marker CHID1 and M1 markers inducible nitric oxide synthase (iNOS) and IDO1. Immunohistochemical analysis revealed granular cytoplasmic expression of CHID1 in all macrophages, both in the tumor stroma and in tumor nests. At the same time, cytoplasmic expression of CHID1 was observed in tumor cells both in adenocarcinoma and in squamous cell carcinoma (Figure 5). Analysis of iNOS and IDO1 expression revealed only a very few iNOS-positive and IDO1-positive TAMs, however, their expression was frequently found in tumor cells. Analysis of PD-L1 was found only in some macrophages but was frequently observed in tumor cells (Figure 5).

Observed expression pattern of CHID1, iNOS, IDO1, and PD-L1 prompted us to analyze the association of the expression of these markers in tumor cells with tumor characteristics (Figure 6).

Analysis of the expression of CHID1, iNOS, IDO1, and PD-L1 in tumor cells was done using immunohistochemistry. Expression of iNOS was found in 74% of samples, expression of CHID1 was found in 100% of cases, expression of IDO1 was found in 60% of samples, and expression of PD-L1 was found in 96% of samples when a cut-off of 1% of positive tumor cells was used. When a cut-off of 10% stained tumor cells was used to define positive samples, 32% of cases were defined as iNOS positive, 25% as IDO1 positive, and 59% as PD-L1 positive. Analysis of association with clinicopathological characteristics of the tumors revealed that high expression of iNOS was observed predominantly in squamous cell carcinoma (*p* = 0.0005), in tumors of central localization (*p* = 0.0128), and in high grade tumors (*p* = 0.0376) (Figure 6). High expression of CHID1 in contrast was found predominantly in adenocarcinoma (*p* < 0.0001) and in tumors of peripheral localization (*p* = 0.0001) (Figure 6). For IDO1 and PD-L1, no association with clinicopathologic characteristics was found.

### 2.5. Survival Analysis

To identify potential prognostic significance of CHID1, iNOS, IDO1, and PD-L1 expression in the patients with NSCLC, the impacts of those markers on the survival was explored.

Analysis of prognostic significance of studied markers revealed that low IDO1 expression in tumor cells indicates poor prognosis in squamous cell lung carcinoma (HR = 2.405, *p* = 0.0467), which corresponds to its immunosuppressor properties. Expression of PD-L1 did not show any prognostic significance in this study.

We established that high expression of iNOS was a marker of favorable prognosis for squamous cell lung carcinoma (HR = 0.3939, *p* = 0.0456), and also NSCLC in general (HR = 0.4418, *p* = 0.0453) (Figure 7). It is also important to mention the prognostic value of iNOS for different stages of the disease. At early stages, the expression of iNOS is not a prognostic factor, while for later stages, high iNOS expression is a marker of good prognosis (HR = 0.376, *p* = 0.0361) (Figure 7).

Analysis of prognostic value of high CHID1 expression showed that it was a marker of good prognosis in adenocarcinoma (HR = 0.3196, *p* = 0.0127) and in NSCLC in general (HR = 0.4019, *p* = 0.0115) (Figure 7). As well, high CHID1 expression was a marker of good prognosis at the early stages of the disease (HR = 0.2111, *p* = 0.0264) (Figure 7).

Observed prognostic significance of both markers for NSCLC in general prompted us to analyze the prognostic significance of a combination of CHID1 and iNOS expression. For this analysis, a high expression group was formed comprising samples that showed both high CHID1 and high iNOS expression. Analysis of prognostic significance of the CHID1/iNOS combination showed HR = 0.2699 (*p* = 0.0129) (Figure 8). It is also important to mention that overall survival of patients from the low CHID1/iNOS expression group was 20 months, while overall survival in the low CHID1 group was 28 months and in the low iNOS group was 50 months.

## 3. Discussion

In NSCLC tumors, as in most other types of solid tumors, inflammatory infiltrate plays an important role in the development and progression of the disease. On the one hand, they can and should participate in the antitumor immune response, but the opposite is often observed, and these cell types, together with tumor cells, suppress the immune response and promote tumor growth. This work is devoted to the analysis of immune components of the microenvironment of NSCLC tumors and their prognostic significance.

Tumor-infiltrating cytotoxic lymphocytes (CD8+) are the main effector cells involved in the antitumor immune response. Depending on the degree of TILs infiltration, all tumors can be classified as cold (low infiltration) or hot tumors (high infiltration), which plays an important prognostic role, especially in combination with modern immunotherapy.

We performed a quantitative analysis of the contents of CD3+, CD8+, and FOXP3 + T cells in 100 samples of NSCLC of various histological types. We showed that a low CD8 + lymphocyte count is a favorable prognostic factor in the early stages of the disease (Figure 1). This is consistent with published data showing that a high CD8 + cell count in NSCLC tumors is a factor in poor prognosis [9]. We also showed that the early stages of NSCLC are characterized by a greater number of both cytotoxic T-lymphocytes and TILs in general (Figure 1). The results suggest that immunosuppression in NSCLC tumors develops in the process of disease progression.

The prognostic significance of FOXP3 + cells infiltrating lung tumors is described in the literature. Most researchers are inclined to believe that a large number of regulatory T cells is an unfavorable prognostic factor [10,11]. However, for some types of tumors, such as esophageal cancer and colorectal cancer (CRC), a large number of FOXP3 + Tregs in tumors may be a marker of good prognosis [12]. In our work, we did not find any prognostic significance of high FOXP3 expression. We also showed that tumor infiltration with regulatory T cells is significantly higher in samples of squamous cell carcinoma than in those of adenocarcinoma (*p* = 0.0005).

Expression of macrophage markers by tumor cells is a highly interesting and important issue. Such markers can be roughly divided into two groups. The first are macrophage markers that are expressed in healthy organisms exclusively in macrophages and the second are expressed in macrophages, but also in some normal tissue cells, especially of epithelial origin. The typical representative of the first group is CD163, the expression of which was reported in various types of tumors. In a healthy organism, it is expressed in monocytes and M2 macrophages and is responsible for haptoglobin scavenging [13]. In cancer, it can be expressed by tumor cells, and its expression in colorectal cancer cells, for instance, is associated with poor prognosis and higher macrophage infiltration [14]. As well, the expression of CD163 was demonstrated in breast cancer cells and in malignant melanoma. This phenomenon is frequently explained by fusion of epithelial cells and myeloid cells, leading to formation of a hybrid cell that bears properties of both macrophage and tumor cells.

We showed that a high content of CD163 + macrophages in a tumor is reliably associated with a poor prognosis, which is generally consistent with the existing concept that strong M2 tumor infiltration by macrophages is a poor prognostic sign for NSCLC [15]. For CD204, which are also referred to as macrophages of the second type, we showed favorable prognostic significance in the early stages of the disease (HR = 0.3187, *p* = 0.0451). In addition, some published studies show that a large number of both CD68 + cells in the tumor and CD204 + macrophages can also be a favorable prognostic factor [16].

IDO1 (indoleamine-2,3-dioxygenase) is an enzyme whose main function is the degradation of tryptophan to form N-formylkynurenine and then a number of products that significantly affect the functions of the immune system. The activated enzyme IDO1 reduces Trp levels in the tumor microenvironment, and this decrease has an intratumoral immunosuppressive effect. One of the main activators of IDO1 expression in a tumor is INF-γ [17]. The main immunosuppressive mechanism for influencing a tumor for IDO1 is a decrease in tumor infiltration by cytotoxic T cells and an increase in the number of regulatory T cells [18]. Since IDO1 is an immunosuppressive component of the tumor storm, it can be assumed that its high expression may be associated with a poor prognosis. Partially for NSCLC, this is confirmed by published data [19], however, more often the expression of IDO1 in NSCLC is not associated with the prognosis [20]. Our studies have shown a poor prognostic significance of high IDO1 expression in squamous cell carcinoma of the lung.

PD-L1 (programmed cell death ligand 1, CD274) is a glycoprotein that controls the activity of the immune system by interacting with the PD-1 receptor on the surface of T cells, inducing apoptosis and inhibition, helping to inhibit the autoimmune response during the inflammation or anti-tumor response. Expression of PD-L1 is observed on macrophages, dendritic cells, B and T lymphocytes, epithelial, and muscle and endothelial cells, while PD-1 is expressed mainly on CD8 + cells [21]. In addition, PD-L1 can often be found in tumor cells [22]. At the same time, the prognostic role of PD-L1 is ambiguous. Various studies have shown that PD-L1 expression is associated with a better prognosis [23,24], a worse prognosis [25], or does not reflect predictive value [26]. Our studies also showed no predictive value of PD-L1 expression in NSCLC.

iNOS is a typical representative of a second group of macrophage markers. It is expressed in macrophages but can be also found in some epithelial cells. iNOS is the main source of NO, the expression of which is increased by various cytokines and growth factors [27]. Nitric oxide, in turn, takes part in the development of various diseases, including oncological diseases, however, increased production of nitric oxide in a tumor can also contribute to antitumor activity through cytotoxic activity. iNOS, by the nature of its activity, can suppress the antitumor immune response. For example, NO is able to inhibit IDO1 both by direct binding and by enhancing its proteosome degradation [28].

Despite a clear association with tumorigenesis, the prognostic role of iNOS is currently not completely determined. Only one study has been published for NSCLC, indicating that iNOS expression in tumors of a given localization can serve as a marker for good prognosis [29], while the results of other localizations are opposite to that [30]. We showed that the high expression of this enzyme in tumor tissue is a marker of a good prognosis for NSCLC and for squamous lung cancer in particular (HR = 0.4418, *p* = 0.0453; HR = 0.3939, *p* = 0.0456, respectively). For adenocarcinomas, this trend is also observed (HR = 0.4131, *p* = 0.1623). What is indicative of the prognostic value of iNOS also depends on the stage of the disease. In the early stages, its expression does not correlate with the prognosis, but in the later stages, it is a favorable prognostic factor (HR = 0.3764, *p* = 0.0361). We have shown that increased iNOS expression is characteristic of squamous cell carcinoma tumor cells, compared with adenocarcinomas.

Chitinase like protein CHID1 seems to belong to this first group of markers too. Expression of CHID1 was demonstrated so far only in macrophages, and its expression is regulated in a manner similar to that of CD163, i.e., its expression is activated by IL-4 and dexamethasone [7]. Expression of CHID1 in pathology was demonstrated for blood monocytes of rheumatoid arthritis patients and in some other inflammatory diseases. In this manuscript, for the first time, we demonstrated the expression of CHID1 in NSCLC cells. The pattern of intracellular CHID1 expression is in good agreement with data published previously [7]. While there is no published data on the role of CHID1 in lung cancer, it was studied in a mouse model of breast cancer. It was demonstrated that forced overexpression of CHID1 in TS/A breast cancer cells led to delayed tumor growth, and this phenomenon was associated with decreased infiltration of the tumors by macrophages [14]. In our study, we also analyzed the density of CD68+ cells in the tumors and found no correlation between CHID1 expression and macrophage density (data not shown). We conclude that more research is needed to explain the antitumor effect of CHID1. Regarding the prognostic value of macrophage markers, there is also a difference between CHID1 and CD163. While the expression of CD163 in tumor cells is a marker of poor prognosis, high CHID1 expression is a marker of favorable prognosis.

It is important to note that when used in combination, the expression of CHID1 and iNOS show higher predictive capacity compared to their individual use (HR of 0.2699 vs. 0.4418 and 0.4019, respectively). This may be an indication of different tumor-suppressing mechanisms of action of CHID1 and iNOS that have an additive effect on the tumor progression.

In summary, for the first time, we demonstrated that chitinase-like protein CHID1 can be expressed by NSCLC cells, and its high expression is a marker of good prognosis for adenocarcinoma and NSCLC in general. At the same time, high expression of iNOS in tumor cells is a marker of good prognosis in squamous cell carcinoma. When used in combination, CHID1 and iNOS show a very good prognostic capacity for NSCLC. Taken together, our data indicate that in the case of lung cancer, tumor-associated macrophages are likely ineffective as therapeutic targets. Sole targeting of a macrophage population in general or the population of type 2 macrophages specifically will not be advantageous to patients. It is important to consider additionally other factors like the tumor microbiome or tumor-specific immune check-point markers. At the same time, macrophage markers expressed by tumor cells may be considered as targets for anti-tumor therapy or, as in the case of CHID1, as potential anti-tumor agents.

## 4. Patients and Methods

### 4.1. Ethics Statement

The Institutional Review Board of N.N. Blokhin Russian Cancer Research Center approved the project, and all patients (approval No. 09/2018) who were involved in the study gave written informed consent that their samples could be used for investigational purposes. Data were analyzed anonymously. All potential participants who declined to participate or otherwise did not participate were eligible for treatment (if applicable) and were not disadvantaged in any way by not participating in the study.

### 4.2. Patients

We collected tissue specimens from 100 patients who received lung cancer resections at the N.N. Blokhin Russian Cancer Research Center. Patients with squamous cell carcinoma (SCC) and adenocarcinoma (AC) were eligible; but patients with small cell lung cancer, sarcoma, or carcinoid tumors were excluded. Eighty patients were followed-up with until it was no longer possible. The median follow-up time was 39.0 months. The formalin-fixed, paraffin-embedded tissue blocks were retrieved from the Pathology Department and sectioned for immunohistochemistry. Clinicopathological data were also collected from patient medical records (Table 1) according to the TNM criteria of the Union for International Cancer Control. The histopathological diagnosis was established for each patient according to the World Health Organization guidelines.

### 4.3. Immunohistochemical Staining

Four- micrometer thick sections were deparaffinized and heated to 110 °C for 10 min for antigen retrieval in ethylenediaminetetraacetic acid (EDTA) buffer, pH 9.0. After cooling, endogenous peroxidase quenching was blocked by 3% hydrogen peroxidase for 5 min in room temperature (RT). Then, the slides were blocked with 5% FBS (RT) for 15 min and incubated for 1 h with primary antibodies: anti-iNOS (SAB5500152; Sigma, St. Louis, MO, USA, 1:150 dilution), anti-CD206 (Sigma-Aldrich, St. Louis, MO, USA, HPA004114), CD204 (Sigma-Aldrich, St. Louis, MO, USA, HPA000272), anti-CD68 (Genemed, South San Francisco, CA, USA, 61-0184), anti-CD163 (Clone 10D6; BIOCARE, Pacheco, CA, USA), anti-CD8 (Genemed, South San Francisco, CA, USA, 61-0124), anti-CD3 (Genemed, South San Francisco, CA, USA, 61-0011), anti PD-L1 (Clone E1L3N, Cell Signaling, Denver, MA, USA), anti-FOXP3 (Cell Signaling, Denver, MA, USA, #98377), and anti-CHID1 (clone PBM-3D4, PrimeBioMed, Moscow, Russia) [31]. Antibody was removed and 100 µL DAB (UltraVision Quanto Detection System HRP DAB, Thermo Fisher Scientific, Waltham, MA, USA) was added to each section. We performed counterstaining with hematoxylin and washed the sections in dH_2_O two times for 5 min each. After dehydration, the sections were mounted with coverslips.

To score the immunostaining results for macrophages (CD68, CD163, CD206, CD204, CHID1) and T-cells (CD3, CD8), we randomly selected five representative high-power microscopic fields (×400 magnification) of the tumor samples per section and counted the numbers of positively-stained cells (Olympus; Tokyo, Japan). Necrotic areas were ignored. The mean percentages of stained cells were counted as 0 (negative), 1 (≤10%), 2 (11–50%), and 3 (>50%). Foxp3 expression was evaluated according to the average number of positively-stained cells in 5 randomly and averagely selected 400× high-power fields (HPF) in each case: 0 (no positive cells), 1 (1–5 positive cells), 2 (6–25 positive cells), and 3 (>25 positive cells) per HPF. Samples with scores 0–1 for CD206, CD8, and FoxP3 were combined in a group with low expression and samples with scores 2–3 were combined in a group with high expression. For CD68, CD163, CHID1, and CD3, samples with scores 0, 1, and 2 were combined in a group with low expression and samples with a score of 3 represented a group with high expression [32,33].

To score the immunostaining results for tumor cells, CHID1 in the cells was classified as positive when clear cytoplasmic staining was present. Since there were no negative cells in the tumors, the intensity of staining was evaluated as 1—weak, 2—moderate, and 3—strong. For the analysis of survival, samples with intensities 2 and 3 were combined in one group.

For iNOS, IDO1, and PD-L1, immunohistochemical staining was scored in tumor cells. Since there are no clinically accepted thresholds for iNOS, IDO1, and PD-L1 expression, the following cutoffs were used for this stain expression: 0—≤1%, 1—1–10%, 2—10–50%, and 3—>50% of tumor cells showing cytoplasmic positivity. For the analysis of survival, samples with scores 0 and 1 were combined in a group with low expression and samples with scores 2 and 3 were combined in a group with high expression, i.e., 10% cut-off.

### 4.4. Statistical Analysis

Statistical calculations were performed using Graph Pad prism 8.3.1 (GraphPad). The Mann–Whitney nonparametric test was used to compare between two groups to examine the association between immune marker expressions and clinicopathological characteristics (stage, grade, histology, location, tumor size, and nodal status). The OS was calculated from the dates of surgery and death from any cause as the endpoint. Survival analysis was estimated by the Kaplan–Meier method, and the logrank test was used to compare survival between the groups. All analyses were two-sided and *p* values < 0.05 were considered statistically significant.

## Figures and Tables

**Figure 1 ijms-22-00450-f001:**
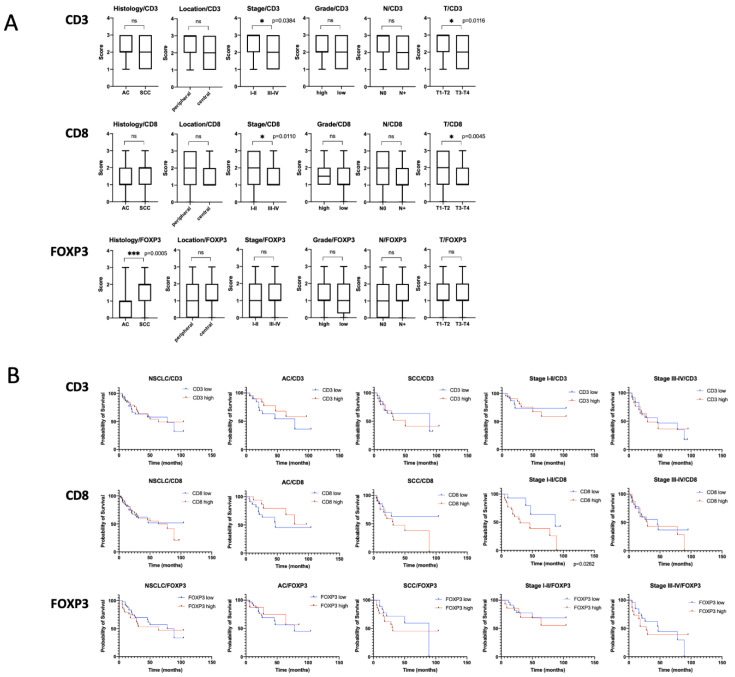
(**A**). Mann–Whitney analysis of CD3, CD8, and FOXP3 expression association with the clinicopathological properties of the analyzed tumors. * indicates *p* < 0.05, *** indicates *p* < 0.001. (**B**). Kaplan–Meier curves of overall survival (OS) in non-small cell lung cancer (NSCLC) based on CD3, CD8, and FOXP3 expression.

**Figure 2 ijms-22-00450-f002:**
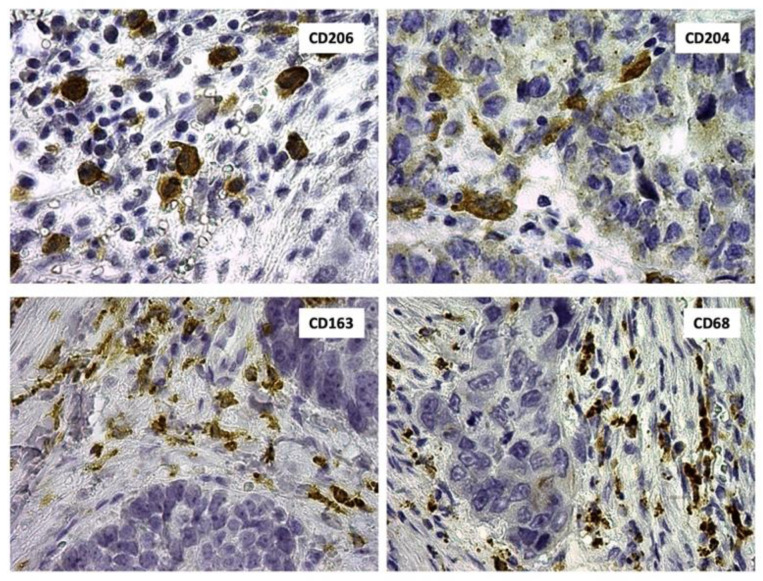
Immunohistochemical analysis of CD206, CD204, CD163, and CD68 in NSCLC samples.

**Figure 3 ijms-22-00450-f003:**
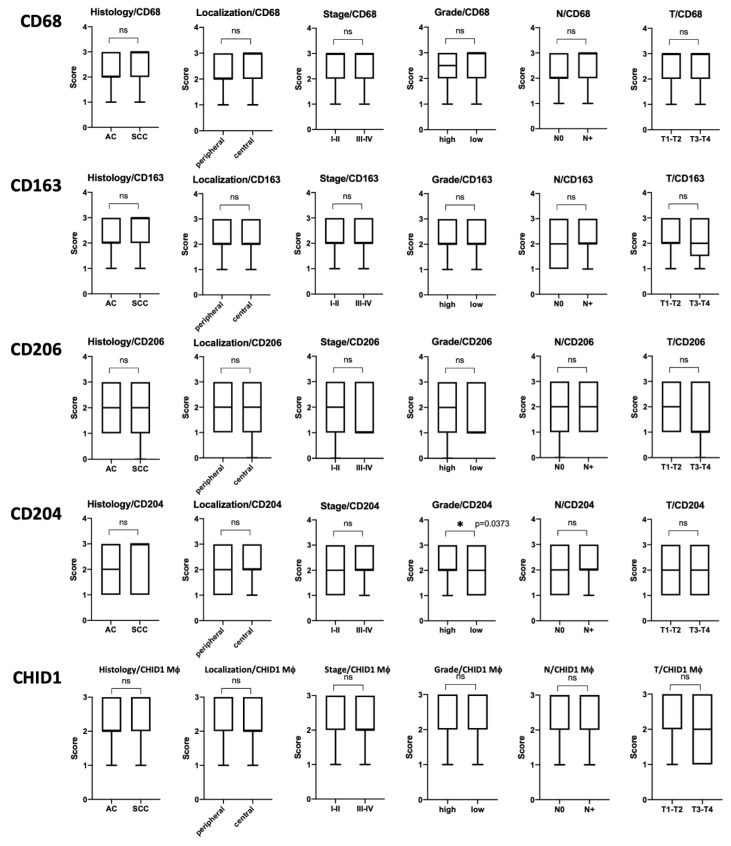
Mann–Whitney analysis of CD68, CD163, CD206, CD204, and CHID1 expression association with clinicopathological properties of the analyzed tumors. * indicates *p* < 0.05.

**Figure 4 ijms-22-00450-f004:**
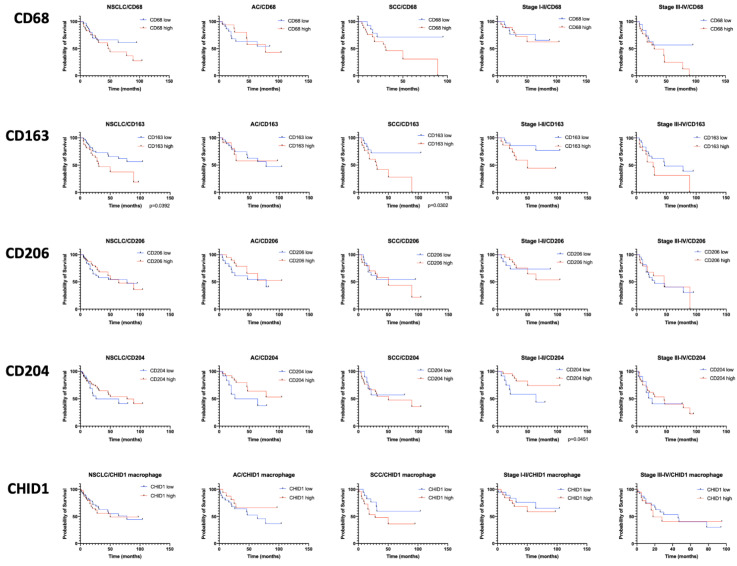
Kaplan–Meier curves of overall survival (OS) in different subgroups of NSCLC based on CD68, CD163, CD206, CD204, and CHID1 expression in macrophages.

**Figure 5 ijms-22-00450-f005:**
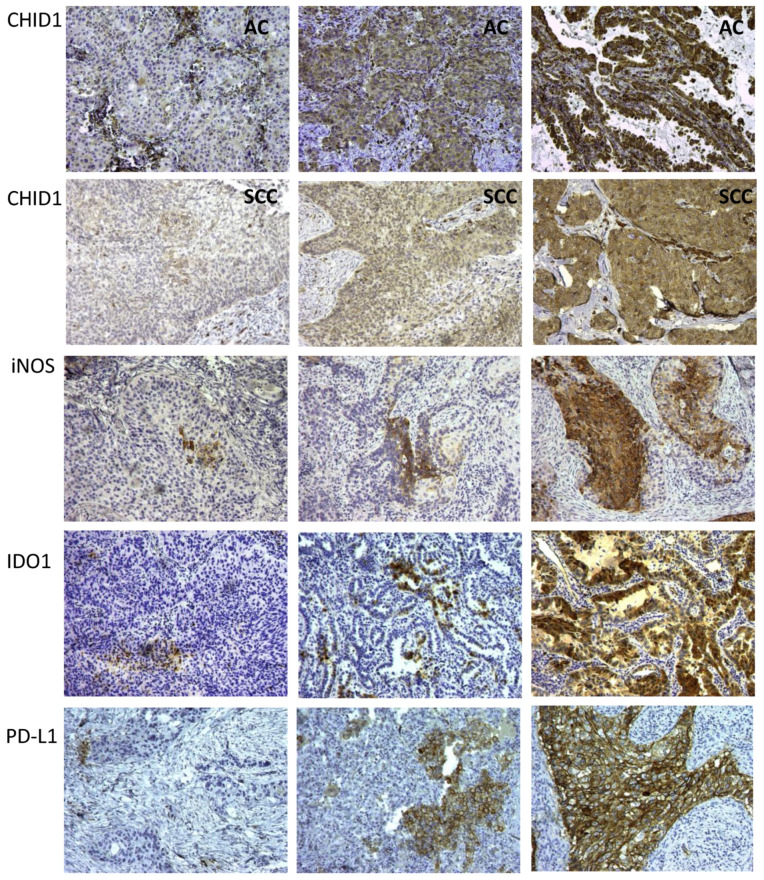
Immunohistochemical analysis of CHID1, iNOS, IDO1, and PD-L1 expression in NSCLC samples. AC—adenocarcinoma, SCC—squamous cell carcinoma. Magnification—100×.

**Figure 6 ijms-22-00450-f006:**
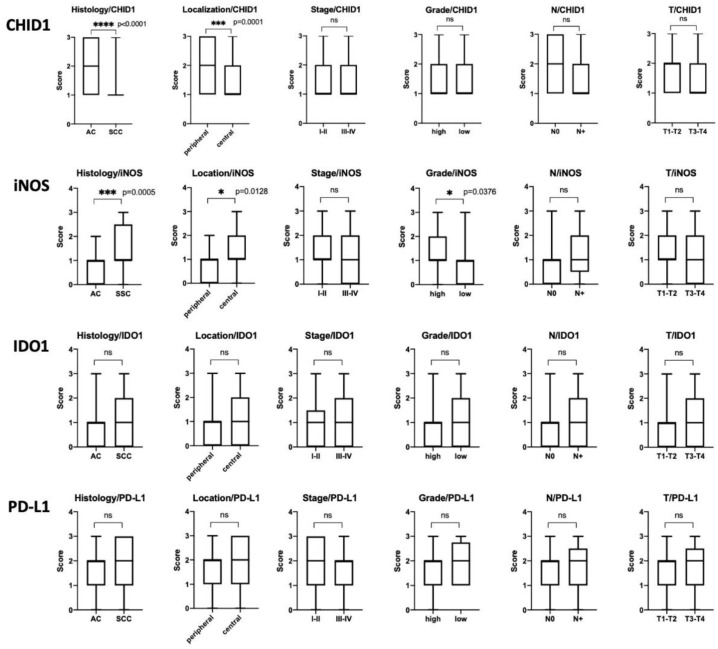
Mann–Whitney analysis of CHID1, iNOS, IDO1 and PD-L1 expression association with clinicopathologic properties of analyzed tumors. * indicates *p* < 0.05, *** indicates *p* < 0.001, **** indicates *p* < 0.0001.

**Figure 7 ijms-22-00450-f007:**
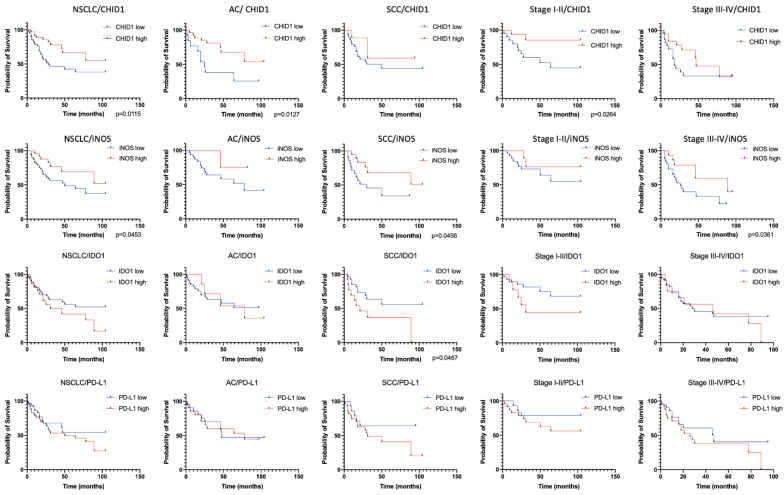
Kaplan–Meier curves of overall survival (OS) in different subgroups of NSCLC based on CHID1, iNOS, IDO1, and PD-L1 expression in tumor cells.

**Figure 8 ijms-22-00450-f008:**
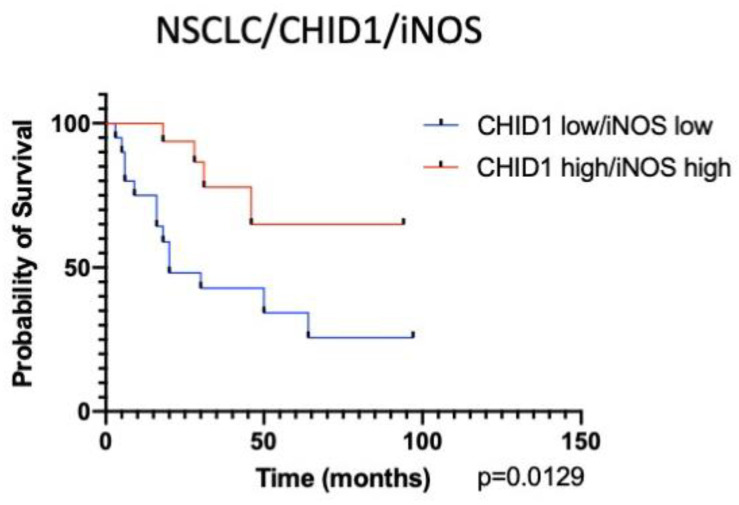
Kaplan–Meier curves of overall survival (OS) of NSCLC based on combined CHID1/iNOS expression analysis.

**Table 1 ijms-22-00450-t001:** Clinicopathological characteristics of 100 patients.

Clinicopathological Parameters	Cases (%)
Age	
<60	40 (40%)
≥60	60 (60%)
Histology	
AC	51 (51%)
SCC	49 (49%)
Stage	
I-II	49 (49%)
III-IV	51 (51%)
Location	
central	42 (42%)
peripheral	58 (58%)
Tumor size (T)	
1–2	55 (55%)
3–4	45 (45%)
Nodal status (N)	
N0	35 (35%)
N+	65 (65%)
Grade (G)	
G1–G2 (high)	60 (60%)
G3–G4 (low)	40 (40%)

## Data Availability

All data was included in the manuscript. No additional deposition of the data was done.

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
