# Peer review of "CHID1 Is a Novel Prognostic Marker of Non-Small Cell Lung Cancer"

_ijms, 2021, doi:10.3390/ijms22010450_

Round 1

Reviewer 1 Report

The authors investigated some promising markers by IHC in a series of NSCLC.

Even if I don't like study simply including some IHC markers in NSCLC to demonstrate some prognostic role (no immunomarker has been included in routine practice despite million of papers on this issues !), the study seems interesting.

The main criticism is related to 1% of cutoff to quote negative and positive cases. This is a by chance cut-off. To demonstrate more robust and reliable data I would expect that a more solid cut-off (i.e., 10%) could be used.

Author Response

We would like to thank the reviewer for positive the evaluation of our manuscript. We removed the definition of positive samples by cut-off 1% from the methods section and added information about the number of positive samples defined by cut-off 10% in the results section (lines 167-169). Please note that for survival analysis groups with cut-off 10% positive tumor cells were used.

Reviewer 2 Report

In this study, the authors examined the association of CHID and iNOS expression in macrophages and tumors with the clinicopathological parameters and overall survival of patients with NSCLC. It was concluded that both CHID1 and iNOS were expressed in NSCLC cells, and high expression of CHID and iNOS were associated with good prognosis for NSCLC. Below are a few questions for the authors to address. 

1. Although the title of the manuscript is "CHID1 is a novel prognostic marker of non-small cell lung cancer", the study showed that the expression of iNOS was also associated with the prognosis of NSCLC to some degree. Was the expression of iNOS correlated with that of CHID1 in tumors? 

2. Line 122- 124. "Immunohistochemical staining for CD68, CD163, CD206, CD204 and CHID1 revealed membrane and cytoplasmic staining of macrophages." Please provide the representative images of IHC staining for CD68, CD163, CD206 and CD204 in the manuscript.

3. Data in Figure 1a and Figure 2 should be presented in a table format to indicate the total number of cases included and the distribution of high or low expression of individual markers in each category. Without those numbers, readers would not be able to verify if the statistical analysis results are correct.

Author Response

We would like to thank the reviewer for the evaluation of out manuscript. We have revised the manuscript according to the comments as follows.

  1. Expression of CHID1 and iNOS do not correlate. Though iNOS seems to be marker of good prognosis as well, we prefer to leave the manuscript title unchanged, since CHID1 as marker of good prognosis is a new finding, while iNOS was already described as a marker of good prognosis of lung tumors.
  2. Representative images were added as figure 2.
  3. Data presented on figures 1a and 2 are provided in addition to the figures in the manuscript as supplementary tables 1 and 2.

Round 2

Reviewer 1 Report

None

Reviewer 2 Report

The authors have addressed all my concerns.